# Combining Blockchain Technology and the Physical Internet to Achieve Triple Bottom Line Sustainability: A Comprehensive Research Agenda for Modern Logistics and Supply Chain Management

**Horst Treiblmaier** 

Department of International Management, MODUL University Vienna, Am Kahlenberg 1, 1190 Vienna, Austria; horst.treiblmaier@modul.ac.at

**Abstract:** Integrating triple bottom line (TBL) goals into supply chains (SCs) is a challenging task which necessitates the careful coordination of numerous stakeholders' individual interests. Recent technological advancements can impact TBL sustainability by changing the design, structure, and management of modern SCs. Blockchain technology enables immutable data records and facilitates a shared data view along the supply chain. The Physical Internet (PI) is an overarching framework that can be applied to create a layered and comprehensive view of the SC. In this conceptual paper, I define and combine these technologies and derive several high-level research areas and research questions (RQ) to investigate adoption and management as well as structural SC issues. I suggest a theory-based research agenda for the years to come that exploits the strengths of rigorous academic research, while remaining relevant for industry. Furthermore, I suggest various well-established theories to tackle the respective research questions and provide specific directions for future research.

**Keywords:** blockchain; distributed ledger technology; physical internet; logistics; supply chain management; research framework; innovation; information technology; triple bottom line; sustainability

---

## 1. Introduction

Information technology (IT) is not only a crucial constituent of modern supply chains (SCs), but also their main innovation driver [1–3]. Consequently, it has played an important role in previous research on SCs and value networks, yielding a wide range of topics such as the usefulness and performance of IT, the use of RFID (radio frequency identification), process design, optimization, and inter-organizational orientation [4,5]. Every technological wave opens up new possibilities, but simultaneously brings along new challenges. Progressive computerization throughout the 1960s and 1970s enabled innovation in logistics planning, ranging from randomized storage to inventory optimization and truck routing. In the 1980s, personal computers emerged, which further improved planning activities, followed by the widespread adoption of networked computing and enterprise resource systems (ERPs) in the 1990s [6]. The proliferation of networking technology based on TCP/IP replaced face-to-face management, manual tracking systems, paper-dominated order processing, and wired communication links with Internet-based purchasing, computer-supported relationship management, and electronic marketplaces [7]. In the early 2000s, for example, a major focus of supply chain management (SCM) research was on the implementation of different IT systems including ERP, product data management (PDM), customer relationship management (CRM), computer-aided design (CAD), supply chain planning (SCP), manufacturing execution systems (MESs), and electronic commerce technology [8].

Technological progress is far from slowing down, and emerging technologies both in the physical and the virtual sphere bear substantial potential for optimizing value networks that span intra- and inter-organizational processes [9]. Two of these technologies include the blockchain, a distributed ledger technology that fundamentally alters the way in which data is stored, accessed, and processed, as well as the PI, a comprehensive SC framework endorsed, among others, by the European Union (EU). While the former transforms data flows, the latter comprises both physical and informational flows. Properly applied and combined, both of them can support the achievement of key supply chain management objectives [10], such as the creation of value, the satisfaction and loyalty of customers, as well as improved profit margins, profitability, and corporate growth for companies [11]. Furthermore, the optimization of intricate value networks through technology aims not only at increasing effectiveness and efficiency at the level of individual corporations, but also at creating sustainable SCs that minimize the use of scarce resources. It is therefore the intelligent combination of physical, informational, and financial flows that enables the achievement of triple bottom line (TBL) sustainability [12].

TBL sustainability has three major areas, which may be conflicting at times: economic development, environmental performance, and social gains [13]. As is shown in Figure 1, those three areas partially overlap and it is only the coordinated interplay of all three that finally leads to sustainability. In logistics and SCM research, this framework has been successfully applied, for example, to analyze the effect of the sustainable supply chain design of social businesses [12], to show that sustainable supplier co-operation has positive effects on firm performance across all three dimensions [14], and to find viable paths for integrating the social dimension with general sustainable SCM theory and practice [15]. Given its inclusive nature, TBL sustainability provides the ideal target system for a comprehensive logistics and SCM framework. Additionally, substantial literature exists on how TBL reporting can be implemented and how widespread it is among companies [16]. It is noteworthy that the distinction between strong and weak sustainability, the latter of which assumes that a substitution of natural capital by man-made capital is sustainable [17], is beyond the focus of this paper.

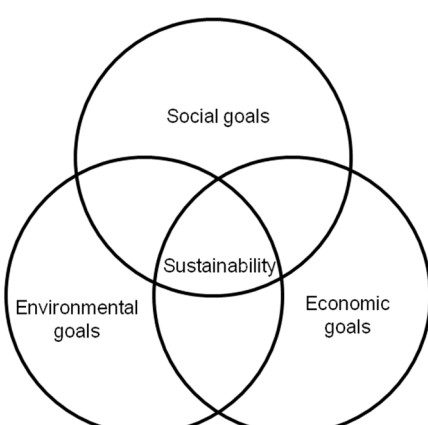

**Figure 1.** Triple bottom line (TBL) goal system.

In the context of logistics and SCM, examples for social goals include the reliable and timely provision of goods to those who need it, the improvement of the working conditions in the transportation industry, and the overall reduction of traffic. The latter is also a highly important environmental goal, since the transport industry accounts for a substantial part of environmental pollution through the emission of pollutants such as $CO_2$, $NO_x$, and fine particles. This situation is aggravated by the transport of empty or half-empty containers and traffic jams especially in urban areas. Furthermore, logistics constitutes a substantial proportion of organizational costs and worldwide GDP and any successful attempt to reduce these costs will yield economic benefits for organizations and nations alike [18–20].

This paper is structured as follows: In Section 2, I briefly elaborate on the basic functioning of the blockchain and the PI, including a short history and the current state of development. In Section 3, I first develop a framework that showcases seven layers of the PI and integrates the blockchain by differentiating between physical, informational, and financial flows. Subsequently, I discuss three different research approaches which can be used to investigate the adoption of novel technologies as well as the resulting structural and managerial challenges. All approaches are based on well-established theories and also include suggestions for action research. I conclude this paper with a brief discussion and a short outlook on future research.

## 2. Blockchain and Physical Internet as Drivers for Logistics and SCM Innovation

Modern supply chains must integrate economic, environmental, and social goals to achieve TBL [13] sustainability. The ongoing quest to coordinate several potentially conflicting goals calls for a flexible supply chain design that integrates the physical chain with information and financial support chains [12]. The advancement of technology supports these efforts. Two of the most prominent technological innovation drivers to emerge during recent years are blockchain technology and the PI, both of which have attracted substantial attention among practitioners and academics.

### 2.1. Blockchain

The blockchain is a "a digital, decentralized and distributed ledger in which transactions are logged and added in chronological order with the goal of creating permanent and tamper-proof records" [21] (p. 547). Its main technological constituents were developed over a period of several decades and include seminal concepts such as linked time stamping, verifiable logs, digital cash, proof of work, byzantine fault tolerance, public key identification, and smart contracts [22]. However, it took the ingenuity of one person (or a group of persons) operating under the pseudonym Satoshi Nakamoto to cleverly combine these technologies to finally solve the problem of double-spending (i.e., the repeated transfer of a digital asset). The cryptocurrency bitcoin was implemented in early 2009 as a working application of blockchain technology [23]. The early years of blockchain research were characterized by scant publicity outside of computer science and cryptography communities and it was not until around 2015 that the general public and businesses began to realize the overall potential of the technology [24–26].

The general functioning of the blockchain has been described in great detail in other publications and is not replicated here [21,23,27]. It is noteworthy, however, that the term blockchain in a strict sense only refers to the concatenation of data blocks containing transactions. The broader term distributed ledger technology (DLT) also includes data structures that do not exhibit a chain-like structure, but can be used for similar purposes as chains of data blocks. Such alternative data structures are often implicitly included in the discussion of blockchain application scenarios and, in this paper, I follow this convention.

From a logistics and SCM perspective, blockchain characteristics such as immutability, transparency, programmability, decentralization, consensus, and distributed trust [28] raise high expectations. Potential use cases are diverse: tracking and tracing, proof of origin, document management, multi-party agreements, payment, integrated financial transactions, and even additive manufacturing. Most interesting for SCM applications, the blockchain allows the transfer of assets without the need for intermediaries, thus increasing end-to-end visibility and speed along the SC. For example, one prominent flagship project launched by IBM and Maersk strives to connect shippers, carriers, customs, and ports, all of which have their own internal data systems [29]. Other well-known examples include Everledger, a startup that stores identifying data on diamonds on the blockchain, the partnership between Walmart, IBM, and Tsinghua University with the goal of tracking the movement of pork in China, and the efforts of the mining company BHP Billiton to track mineral analyses performed by outside vendors [30].

The blockchain can be used to alleviate risks inherent to SCM such as uncertainty regarding quantities of production, lack of transparency when a manufacturer changes suppliers, unethical behavior of middlemen, and complicated inventory management. Taken together, previous research has postulated that the blockchain can positively impact SC sustainability by providing product information to consumers, assuring information veracity, and enforcing representation through smart contracts [31]. However, several potential risks also come along with blockchain implementation, resulting from increased transparency, the immutability of stored records, the necessity for wide adoption and standard implementation, as well as the potential rewriting of transaction history by a single powerful entity [32].

### 2.2. Physical Internet

The goal of the PI is the creation of a global logistics system that is based on the interconnection of existing logistics networks. In order to achieve this, a standardized set of protocols, modular containers, and smart interfaces are applied [33]. Following the descriptions of [18,33,34], it can be defined as follows: *The Physical Internet (PI) is a comprehensive and measurable supply chain framework which is based on a network of physical components. These components are standardized as well as optimized and exchange information to improve the effectiveness, efficiency, and sustainability of supply chain management operations.*

The PI was originally developed by Montreuil [18] who spotted 13 unsustainable characteristics of logistics and SCM, namely, (1) poor space utilization for road, rail, sea, and air transport; (2) empty travel being the norm rather than the exception; (3) bad working conditions for truck drivers; (4) products sitting idle; (5) poor use of production and storage facilities; (6) products being never sold and never used; (7) products not reaching their intended destination; (8) high inefficiencies in multi-modal transportation; (9) dysfunctional city logistics; (10) inefficient cross-docking operations; (11) low network security and robustness; (12) difficulty to justify use of smart automation and IT in supply chains; and (13) limited innovation opportunities. In order to remedy these symptoms and to create sustainable and resilient supply chains, various technical and non-technical issues have to be solved, including changes in organizational structures [35], the selection of the most appropriate level of (de)centralization [36], and the establishment of synchromodal transportation networks that are based on mutual trust, sophisticated planning, and collaboration [37].

The PI not only constitutes a suitable framework for academic research by structuring the research domain, providing goals and a measurement system, but has also been adopted by the European Technology Platform ALICE (Alliance for Logistics Innovation through Collaboration in Europe) to guide its innovation strategy until the year 2050. This Platform was set up to "develop a comprehensive strategy for research, innovation and market deployment of logistics and SCM innovation in Europe" and advises the allocation of EU research funding [38]. Five different working groups have been established to work on "IS for interconnected logistics", "sustainable, safe and secure supply chains", "corridors, hubs and synchromodality", "global supply network coordination and collaboration", and "urban logistics" [38]. This task-oriented work organization fits the layered structure of the PI, which I will detail in the following section.

## 3. Research Agenda

In order to create a comprehensive research agenda, a clear structure is needed and, in the case of academic research, a theoretical underpinning in combination with adequate research questions (RQ) must be deduced. In the following sections, I will develop a high-level agenda in this stepwise manner.

### 3.1. A Layered Framework for Logistics and Supply Chain Management Research

Figure 2 presents a layered framework that integrates the PI with the blockchain. The seven layers of the model are based on Treiblmaier et al. [34]. The lowest layer, modular containers, deals with the creation of modular, ecologically sound, robust, lightweight, and scalable containers. The next layer is about vehicle usage and optimization, which includes sharing of transportation means, ensuring

full loading, energy efficiency, and the use of relays. The following layer deals with the creation of modern and open transit centers that offer a fully functional design and provide effective and efficient cargo handling. Seamless, secure, and confidential data exchange can be achieved by open, shared, and secure protocols as well as mechanisms to regulate data access. A legal framework is needed to ensure legal security, which is especially important in cross-border transport. Cooperation models deal with the equitable sharing of revenues and, finally, specialized business models can be built that fully capitalize on the strengths of the PI resulting from previous layers.

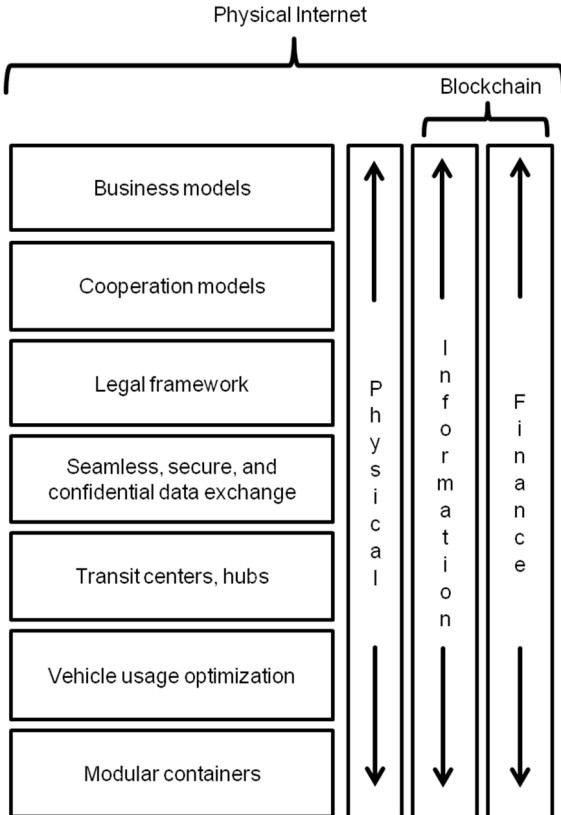

**Figure 2.** A layered logistics and supply chain management (SCM) research framework.

On the right side of Figure 2, the physical, informational and financial flows are shown. The physical supply chain is made up of the flow of goods, while information and finance flows play key supporting roles [39]. The flow channels span all PI layers and play an important role on each respective layer, as well as in connecting different layers. The blockchain affects both non-physical layers, namely, the flow of non-financial information and financial information in the form of payments. Figure 2 thus presents a framework that integrates different perspectives on logistics and SCM and the accompanying flows. It is noteworthy that, similar to the Internet protocol TCP/IP, improvements in each layer can be made without any need to consider adjacent layers, as long as the respective interfaces are taken into consideration.

*3.2. Theoretical Perspectives on Blockchain and PI*

SCM research can benefit from a theoretical approach, since this creates structure and thereby fosters future incremental research. This is true for the SCM domain as a whole [39] as well as for the application of theories from other domains to better understand the functioning of logistics and SCM and their broader implications. Since logistics and SCM research borders on and overlaps with fields as diverse as production planning, management, and economics, it is not surprising that this diversity is also reflected in the types of theories that are applied in academic research. In this paper, the focus is

mainly on those theories that deal with innovation adoption, organizational structures, and managerial issues, especially from a new institutional economics perspective [40]. Using complementary theories not only advances the understanding of SCM, but also helps to motivate further theory development in SCM [41].

Gregor differentiates between different types of theories according to their respective goals: (1) analysis and description, (2) explanation, (3) prediction, and (4) prescription [42]. In this paper, I will mainly elaborate on research questions for explanation and prediction, but I will also suggest research topics that focus on the creation of artifacts, which, according to the proposed framework, can be anything from designing a new modular container to proposing a blockchain-based business model that allows for seamless tracking and tracing of dangerous goods. I will differentiate between two theoretical streams of research that are popular across a wide array of disciplines, namely, diffusion of innovation [43] and technology adoption [44] on the one hand, and the new institutional economics perspective on the other hand. Additionally, I include (adaptive) structuration theory, which has been shown to provide interesting insights for SC research at the firm level, especially when it comes to understanding the difficulties in using IT systems to initiate systemic change [45].

*3.3. Research Areas and Research Questions*

In the following sections, I briefly outline core research areas and corresponding research questions to provide guidance for future studies. All questions are presented at a relatively high level of abstraction and it is up to individual researchers to further specify and operationalize them. In other words, many interesting ideas can be found between the lines and the respective research questions can and should be refined as the technologies improve and new research results emerge, on which further research can be built. Similarly, the suggested theories can be modified as needed and, at times, even the application of a single theoretical approach might suffice [41]. The research questions are clustered in blocks, starting with explanation/prediction and followed by prescription, the latter of which I have labeled as action research in order to highlight the fact that these activities are directed toward practical outcomes and characterized by interactions between researchers, subjects, and the context [46]. I will start with research into adoption, followed by structural and managerial topics.

Figure 3 gives an overview of the theoretical perspectives I present in this paper as well as the respective research areas. It also implies a (research) timeline indicating that, at first, broad questions regarding the antecedents or consequences of the blockchain or the PI must be clarified, followed by a restructuring process that involves individual organizations and the relationships between them and, finally, questions pertaining to the management of these newly created structures. The suggested theoretical approaches enable the generation of insights from different perspectives by building upon existing literature, and allow for the positioning of individual research results within a broader context.

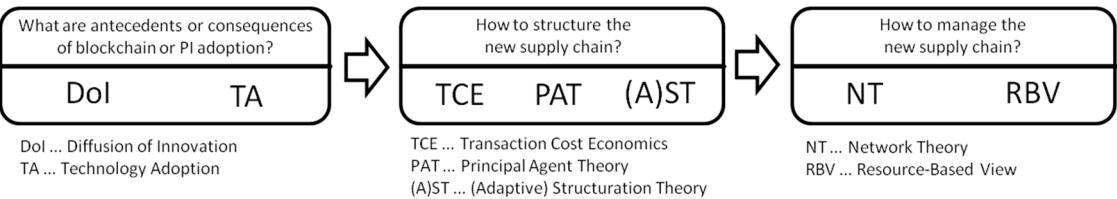

**Figure 3.** Theoretical perspectives and emerging research areas.

3.3.1. Diffusion of Innovation and Technology Adoption Research

Systematic research on the diffusion of innovation (DoI) started in the 1960s with the synthesis of a multitude of diffusion studies by Everett Rogers and the identification of key elements (innovation, adopters, communication channels, time, social system), innovation characteristics (relative advantage, compatibility, complexity, trialability, observability), stages of the adoption process (knowledge, persuasion, decision, implementation, confirmation), and adopter categories (innovators, early

adopters, early majority, late majority, laggards) [47]. Many concepts from DoI research were later combined with behavioral research (e.g., theory of reasoned action (TRA), theory of planned behavior (TPB)) and resulted in a multitude of parsimonious (e.g., technology acceptance model (TAM)) or rather complex models (unified theory of acceptance and use of technology (UTAUT)), which investigate technology adoption (TA) from various angles, considering a wide variety of antecedents as well as numerous mediating and moderating variables. The theoretical underpinnings of these models are extensively discussed in the academic literature across several domains and shall not be reiterated here [48]. Accordingly, previous research has already suggested the application of concepts from DoI or TAM to investigate blockchain [43,49,50] and PI adoption [51]. Care has to be taken, however, not to blindly apply the TAM or one of its many derivations without taking into account the idiosyncrasies of the blockchain or the PI. I therefore suggest the differentiation of two different adoption research streams, the first one focusing on the characteristics of the respective technologies and the second one on the characteristics of the adopting organizations.

The main characteristics of the blockchain are immutability, transparency, programmability, decentralization, consensus, and distributed trust [28]. Introducing such a technology into a company not only poses a major challenge during the implementation phase, but frequently also requires significant structural and procedural changes. Similarly, the characteristics of the PI can be directly derived from Figure 2 and comprise modular containers, universal interconnectivity, container handling and storage systems, smart networked containers embedding smart objects, distributed multi-segment intermodal transport, a unified multi-tier conceptual framework, an open global supply web, product design for containerization, product materialization near to point of use, open performance monitoring and capability certification, reliable and resilient networks, business model innovation, and open infrastructural innovation [18]. The PI is an integrative concept, rather than a single technology, which spans the boundaries between companies and therefore also necessitates substantial changes within and between organizations. Accordingly, the respective characteristics have to be taken into account as a starting point for adoption research:

- **RQ-TA 1a.** How do blockchain characteristics affect organizational adoption?
- **RQ-TA 1b.** How do PI characteristics affect organizational adoption?

Innovation adoption regularly happens within a context that is shaped by a wide variety of organizational characteristics. These can be defined as "features originating both from the management model adopted by the organization, through its structure or strategy, and from the company culture embodied in the nature of its membership and relationships" [52] (p. 22). Additionally, existing inter-organizational relationships might impact the decision-making scope of an organization [53]. It is therefore intra- and inter-organizational characteristics that influence the successful adoption of the PI as an overarching concept and the blockchain as a novel paradigm for the exchange and storage of financial and non-financial information. Future research must identify those characteristics that play a major role in the adoption decision process as well as their level of impact:

- **RQ-TA 2a.** How do organizational characteristics affect blockchain adoption?
- **RQ-TA 2b.** How do organizational characteristics affect PI adoption?

As is shown in Figure 1, TBL sustainability presents a comprehensive goal system that comprises economic, environmental, and social aspects. All three aspects contain a wide selection of subgoals that depend on the respective organization and its context. Organizational logic can be used to prioritize between the different elements of TBL and to create a goal system that follows a clear structure and potentially even a hierarchical order [54]. For example, in their four case studies on sustainable supply chain use among social businesses in Haiti, Bals and Tate identified the following outcomes: financial support for customers, monetary flows back to the community (economic), closed-loop waste management (environmental), better educational outcomes, jobs for disadvantaged groups, funding of social projects, and community pride/commitment (social) [12]. Both the PI and the blockchain have

already been scrutinized regarding their potential to achieve high-level goals in all three dimensions of TBL. The PI was designed such as to increase the overall sustainability of SCs and inherently contains many economic, environmental, and social goals [55]. The blockchain is not only one of the major enablers of the PI, but also has the potential to realize benefits such as economic inclusion or a more sustainable economy [17,56]. An application of a TBL-based sustainability approach therefore implies (a) the identification and operationalization of a goal system with relevant subgoals, and (b) the identification of suitable measures. Two broad research questions emerge:

- **RQ-TA 3a.** How does blockchain adoption impact TBL sustainability?
- **RQ-TA 3b.** How does PI adoption impact TBL sustainability?

Action research, which was developed and described by MIT professor Kurt Lewin and the Tavistock Institute, advocates research activities with practical impact. By following an action research approach, the practical implications of the research project are taken into account and success or failure represents an important part of the project [57]. Given that the PI and the blockchain are currently in a nascent stage of development, and that both of them strive to achieve goals of improving the quality of human lives, an action research approach is therefore frequently justified. The theoretical background can be provided, for example, by the design science research paradigm, which frequently contains prescriptive elements that outline how to best create a specific artifact [42]. Given that the PI is the current framework for EU research funding activities, it is not surprising that the practical relevance of numerous research activities plays an important role and that the majority of PI-related research projects are conducted in close cooperation with industry. Many research studies therefore describe or design practical solutions ranging from the development of modular containers to novel business models that foster inter-organizational cooperation and help to avoid unnecessary transportation [34].

Similarly, many academics investigate the design and implementation of blockchain-based systems that help to solve pending logistics and SCM problems. Examples include the application of blockchain technology to build secure and trusted environments for intelligent vehicle communication [58] or the provision of secure key management between heterogeneous networks to enable intelligent transportation systems [59]. Using the blockchain as a core element within the general PI framework was previously suggested by Galvez and Dallari [60], who developed a shipment use case using Hyperledger Fabric. In this use case, the blockchain supports three transactions that are embedded into the general PI-framework: auctions, exports and payouts, and custody. The authors concluded that a blockchain-based PI infrastructure helps to address economic, environmental, and social sustainability challenges in scalable, modular, and autonomous ways. Following an action research perspective that aims at TBL sustainability, two major research questions can be derived:

- **RQ-TA 4a.** How can the blockchain be implemented to foster TBL sustainability?
- **RQ-TA 4b.** How can the PI be implemented to foster TBL sustainability?

3.3.2. Structural Research

Halldórsson et al. [40] suggested a new institutional economic perspective to investigate topics such as third-party logistics and new product development within SCM. Their framework has previously been adapted to blockchain-related questions [21]. Transaction cost economics (TCE) and principal agent theory (PAT) can be used to answer questions pertaining to the structure of SCs and to investigate them from different angles. TCE especially helps to answer the fundamental question of why firms exist and provides guidance regarding appropriate governance structures. The blockchain is expected to significantly alter intra-organizational structures (e.g., by removing levels of middle management) and inter-organizational structures (e.g., by removing market intermediaries). The PI provides the general framework by specifying layered instances and a communication model between them, called an open logistics interconnection (OLI) model. Following the ISO/OSI reference model for open systems interconnection, the OLI has seven layers (i.e., physical layer, link layer, network layer, routing layer, shipping layer, encapsulation layer, logistics web layer), which receive

services from their lower layers and provide services to their upper layers [61]. All changes in the flow of goods and information induce a change in transaction costs that can be investigated using TCE.

PAT focuses on the relationship between a principal and an agent, both of which act in self-interest, but only the latter possesses all the relevant information. The transparency and immutability of the blockchain alters information access and therefore reduces information asymmetry. For example, integrated supply chain ledgers in permissioned blockchains enable trading partners to post on each other's ledgers [62]. Again, the PI provides the broader framework and allows for an assessment of how SC innovations trigger intra- and inter-organizational structural changes. Related previous research has investigated the structural impact of the PI in numerous areas ranging from the design of vendor-managed inventories based on shared facilities and means of transport [63] to PI networks that span 26 European cities in 9 countries [64].

A third option to investigate intra- and inter-organizational structural changes, which goes beyond the framework of new institutional economics, is structuration theory (ST). ST helps to understand how social systems are produced through social interactions. Based on the work of Giddens [65], it was later developed into adaptive structuration theory (AST) by DeSanctis and Poole, who studied the interaction of groups and organizations with information technology [66]. ST has already been recognized as a valuable framework for logistics and SCM research [67]. In that study, the authors used ST to discuss the adoption of the global transportation network by the US Department of Defense. AST provides new insights into SC research at the firm level, especially when it comes to investigating the difficulties of using IT systems to drive systemic change [45]. Since the blockchain and the PI presuppose the introduction of novel technology, the application of (A)ST might therefore lead to novel perspectives that especially consider social aspects of IT usage.

Given the three-fold sustainability goal structure of the proposed research framework, all structural changes also must be assessed with regard to their economic, environmental, and social implications. This brings forth the following two research questions:

- **RQ-ST 1a.** How does blockchain implementation impact the structure of the SC?
- **RQ-ST 1b.** How does PI implementation impact the structure of the SC?

Action research strives to design and implement intra- and inter-organizational structures that help to create value chains which are TBL sustainable. For example, one of the outstanding features of the blockchain is the possibility to allow shared information access. This enables the creation of structures in which mutual interpersonal trust is replaced by trust in the system. Consequently, various layers of intermediaries whose main tasks are to compile and process information might be replaced or reorganized, which not only lowers total costs, but also optimizes delivery thus reducing emissions and improving quality of life in various ways. More generally, blockchain-enabled solutions such as smart contracts can lead to improved auditability, immutability, and disintermediation, which in turn can help to solve salient SC pain points: traceability, compliance, accountability, and enforcement [31]. Similarly, simulations of PI implementation have illustrated its potential to improve value networks thus leading to decreased traffic and emissions and hence cost and energy savings [68]. Again, the overall goal is to create structural changes which explicitly consider economic, environmental, and social implications. This leads to the following research questions:

- **RQ-ST 2a.** How can blockchain implementation create SC structures that are TBL sustainable?
- **RQ-ST 2b.** How can PI implementation create SC structures that are TBL sustainable?

### 3.3.3. Management Research

Within the comprehensive new institutional economics framework, network theory (NT) and the resource-based view (RBV) can be applied to answer questions pertaining to the management of organizational structures [21,40]. Every comprehensive investigation of the blockchain and the PI must also include managerial issues, since changes in the organizational structure or inter-organizational

relationships are likely to impact management. The focus of NT is on the nature and quality of business relationships, which will presumably be altered by technological advances [69]. The blockchain can be seen as an enabler for increased transparency of information and trust (e.g., through the use of smart contracts) [21], and the PI demands an improved information flow between different entities in the SC as a crucial prerequisite of value chain efficiency. RBV investigates the importance of organizational resources and how they can be used to stay competitive. The blockchain constitutes a technology that can help to save costs and increase informational transparency by streamlining processes and removing intermediaries, whereas the PI provides the broad framework to scrutinize all SC-related technologies and processes. Both of them are likely to impact managerial issues and lead to the following questions which must be investigated under consideration of TBL sustainability:

- **RQ-MA 1a.** How does blockchain implementation impact SC management?
- **RQ-MA 1b.** How does PI implementation impact SC management?

The hype around the blockchain has created a significant amount of interest especially among C-level executives [70]. To a lesser extent, this is also true for the PI, not least because the latter is propagated by the EU and several PI research projects have already been carried out jointly by academia and industry [71]. The blockchain is mainly about the flow of information, while the PI also includes the flow of goods. In combination, blockchain and PI strive to increase the speed and the transparency of these flows. The goal of design-oriented research from a managerial perspective is the design of systems that ensure data security and privacy while simultaneously providing management with the needed information in an aggregated manner. Further implications might include a shift in responsibilities as well as the emergence of completely new skill sets. Research is needed on how management can cope with the challenges of these new technologies:

- **RQ-MA 2a.** How can blockchain implementation support SC management?
- **RQ-MA 2b.** How can PI implementation support SC management?

## 4. Discussion, Conclusions, and Future Research

In this paper, I define and describe the blockchain and the PI and integrate them into an overarching framework for logistics and SCM that has a layered structure and can guide future research and industry projects. Additionally, I derived research questions in various areas that deserve further attention.

Academics can use this framework to study either the respective layers or their interaction. From an industry perspective, the final goal is to create SCs that achieve TBL sustainability by carefully balancing economic, environmental, and social interests [12]. In order to accomplish this, the PI can be used to gather new insights on how to structure and design complete SCs starting from the design of modular containers to the creation of innovative business models [34]. The blockchain represents a promising technology to redesign informational and financial flows, both of which supplement physical flows in a supply chain. The framework presented in this paper integrates the different layers of the PI with those three flows.

In order to foster structured and incremental academic research, I provide three theoretical perspectives that are suitable for studying technology-induced changes from various angles. These perspectives cover technology adoption as well as structural and managerial consequences of technology-induced change and yield a wide range of research questions. Academics benefit from theory-based research which enables a better understanding of why and how novel technologies are adopted, how they imply structural and managerial changes, and in which ways they can contribute to achieve TBL sustainability. Practitioners benefit from a better understanding of the connection between the blockchain and the PI.

I suggest several research areas and high-level research questions that can be directly derived from the underlying theories. Further research is needed to refine these questions and to adapt them to specific research problems.

**Funding:** This research received no external funding.

**Conflicts of Interest:** The author declares no conflict of interest.

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
