# Peer review of "Combining Blockchain Technology and the Physical Internet to Achieve Triple Bottom Line Sustainability: A Comprehensive Research Agenda for Modern Logistics and Supply Chain Management"

_logistics_

Round 1
Reviewer 1 Report
This is an interesting combination of topics, one that is bound to be of high interest to readers of this journal. Blockchain is a hot topic, although not the panacea that some seem to imagine. However, this article also links it to the physical internet. That link alone would be enough to make it a significant contribution to the literature.
Author Response
Response to Reviewer 1 comments
Point 1: This is an interesting combination of topics, one that is bound to be of high interest to readers of this journal. Blockchain is a hot topic, although not the panacea that some seem to imagine. However, this article also links it to the physical internet. That link alone would be enough to make it a significant contribution to the literature.
Response 1: Thank you very much for the positive assessment of my paper. I fully agree with you that Blockchain is not a panacea and I hope that my paper will encourage colleagues from academia to conduct objective and rigorous research in order to find out how Blockchain can actually benefit the industry.
Best regards,
Horst Treiblmaier

Reviewer 2 Report
This paper proposes a comprehensive research agenda for SCM based on two technical advances—blockchain and the physical internet. The topic is well advised, however, I think the paper could be strengthened to make it easier to read and useful to potential readers/researchers. In particular, if the intent of the paper is to offer a research agenda, the resulting proposed research directions and a path forward could be more clearly articulated by a less high-level perspective.
My comments below concern the construction and flow of the paper. The transition between the sections could be better developed. The first sections of the paper are largely descriptive and a tightening of the language would tighten the flow of the paper. The last sections of the paper offer the research questions. In the end, the author(s) could clearly state the most important take-away’s of the paper. The conclusion is a statement of what was done, but not substantive statements of insight or the utility of the framework.
There are important ideas in this paper, but they are obscured by the paper’s organization and writing style. In particular, the paper should be revised so that it is not written in the first person “I”.
Some specific comments
Section 1
Line 32….information security risk and privacy are raised in the 3rd sentence but not developed.
Line 40. Early 2000s instead of 2003?
Line 59. Seven-layer framework is in Section 3.2
Line 62. Eliminate “thoroughly and incrementally”. The reader is the judge of the former, and the later is not highly developed in the paper. Is this prescriptive? Why is incremental important and from whose perspective?
Section 2
Since TBL is a major theme of the paper/title, perhaps should be woven into the introduction. The first mention, other than the title is in this section.
The discussion of blockchain could be better directed to the issues relevant to SCM and the goals of the paper. The description is largely descriptive, as one would find in Wikipedia. The third paragraph of this section comes closest to the themes of the paper, addressing current applications, but further discussion that foreshadows the research questions would be helpful.
Mention of Bitcoin (Lines 80-86) is not critical to the arguments of the paper. Most people reading this paper will know this history. Similar comment for about relating PI to arguments of the paper. For example, follow up sentence “The PI provides a suitable framework for academic research” with why and how related to framework being proposed.
Section 3
Nice statement of agenda
Section 3.1 Include definition of all goals…only social is singled out. Consider moving the discussion of TBL earlier in the paper, perhaps after the third paragraph in the introduction.
Section 3.3. The statements in this section might be more effectively integrated into the first part of Section 3.4 to support the description of the figure on theoretical perspectives.
Many terms are introduced without definition or example…artifact, action research, blocks, etc. and it is confusing to the reader. The reader would benefit from some clarity about what’s what.
Sections 3.4.1, 3.4.2, and 3.4.3
The introduction and discussion of relevant theoretical approaches from other domains is interesting.
Make clearer how blockchain and PI are “integrated in the context of the paper”, per statement in abstract. Some examples bring together blockchain and PI. But the hypotheses are all specific to one or the other technology.
As the authors note, the research questions are all written at a high level of abstraction, leaving it to individual researchers to further specify and operationalize them. For this reviewer, the paper would be more interesting and useful if the authors reached for a lower level of abstraction.
Author Response
Response to Reviewer 2 comments
This paper proposes a comprehensive research agenda for SCM based on two technical advances—blockchain and the physical internet. The topic is well advised, however, I think the paper could be strengthened to make it easier to read and useful to potential readers/researchers. In particular, if the intent of the paper is to offer a research agenda, the resulting proposed research directions and a path forward could be more clearly articulated by a less high-level perspective.
Dear Reviewer,
thank you very much for taking the time to carefully review my paper and to provide valuable feedback. I will explain in the following sections how I have used your feedback to improve my paper.
My comments below concern the construction and flow of the paper. The transition between the sections could be better developed. The first sections of the paper are largely descriptive and a tightening of the language would tighten the flow of the paper. The last sections of the paper offer the research questions. In the end, the author(s) could clearly state the most important take-away’s of the paper. The conclusion is a statement of what was done, but not substantive statements of insight or the utility of the framework.
I have carefully reread the paper and asked a native speaker to critically review it. We have made some minor changes in order to improve the flow. Additionally, I revised the conclusion a bit in order to better highlight the utility of the framework. In a nutshell I believe that this paper will mainly be useful for academics who are looking out for interesting research questions.
There are important ideas in this paper, but they are obscured by the paper’s organization and writing style. In particular, the paper should be revised so that it is not written in the first person “I”.
I know that there are different opinions in the academic community on what constitutes a good writing style. Admittedly, I have preferred passive voice over active voice for many years myself, since this was the way it was taught in school or at university. However, I have changed my opinion based on several discussions with linguists. Nowadays, many journals even advocate active voice, since it produces “clearer and more succinct sentences” (JAIS style guide, p. 1). Please find here the official style guide of the Journal of the Association for Information Systems:
In the past I have frequently used plural for papers that I have authored alone, but one reviewer once pointed out that this sounds weird. I have therefore purposefully decided in this paper to use active voice in combination with singular and hope that you find this acceptable.
Some specific comments
Section 1
Line 32….information security risk and privacy are raised in the 3rd sentence but not developed.
You are absolutely right. There exists a plethora of literature on Blockchain and security/privacy, but in this paper I do not want to go into that level of detail. I therefore decided to remove this statement.
Line 40. Early 2000s instead of 2003?
I took the year 2003 from the literature. I fully agree with you that “early 2000s” not only sounds better, but is also more appropriate and I changed the text accordingly.
Line 59. Seven-layer framework is in Section 3.2
I have corrected that. Thank you for reading my paper so carefully.
Line 62. Eliminate “thoroughly and incrementally”. The reader is the judge of the former, and the later is not highly developed in the paper. Is this prescriptive? Why is incremental important and from whose perspective?
I agree and I removed that statement. “Incremental” was indeed an inappropriate term. I was referring to the layered nature of the framework but this not necessarily require an incremental research design.
Section 2
Since TBL is a major theme of the paper/title, perhaps should be woven into the introduction. The first mention, other than the title is in this section.
TBL is a crucial concept and I agree with you that it definitely should be mentioned in the introduction. Please note that I highlight the importance of sustainability (and the concept of TBL) in the second paragraph of the introduction, followed by the discussion of sustainability.
The discussion of blockchain could be better directed to the issues relevant to SCM and the goals of the paper. The description is largely descriptive, as one would find in Wikipedia. The third paragraph of this section comes closest to the themes of the paper, addressing current applications, but further discussion that foreshadows the research questions would be helpful.
I agree with you that the Blockchain should not be discussed in detail in this paper, which is why I have reduced this discussion to a single paragraph. I therefore just included a brief definition and a very short explanation.
Mention of Bitcoin (Lines 80-86) is not critical to the arguments of the paper. Most people reading this paper will know this history. Similar comment for about relating PI to arguments of the paper. For example, follow up sentence “The PI provides a suitable framework for academic research” with why and how related to framework being proposed.
I have shortened the Bitcoin section and now only mention the term once as the first functioning implementation of a Blockchain. I have also clarified the advantage of the PI as a comprehensive framework for academic research.
Section 3
Nice statement of agenda
Thank you!
Section 3.1 Include definition of all goals…only social is singled out. Consider moving the discussion of TBL earlier in the paper, perhaps after the third paragraph in the introduction.
Following your advice, I decided to rearrange the structure of the paper and to move the discussion of TBL into the introduction section.
As far as the “definitions” are concerned, I did not want to provide a definition for any of the three sustainability types, since this is far beyond the scope of this paper. I rewrote this statement such that it is hopefully clear now that these were just some examples for social costs that are influenced by SCM / logistics. Economic and environmental costs should be much more intuitive, but I also included some examples for those.
Section 3.3. The statements in this section might be more effectively integrated into the first part of Section 3.4 to support the description of the figure on theoretical perspectives.
Many terms are introduced without definition or example…artifact, action research, blocks, etc. and it is confusing to the reader. The reader would benefit from some clarity about what’s what.
Thank you for pointing this out. I included examples for artifacts the first time the term was mentioned and explained the term "action research". I hope that by using the term “data blocks” it is understandable what I am referring to.
Sections 3.4.1, 3.4.2, and 3.4.3
The introduction and discussion of relevant theoretical approaches from other domains is interesting.
Thank you!
Make clearer how blockchain and PI are “integrated in the context of the paper”, per statement in abstract. Some examples bring together blockchain and PI. But the hypotheses are all specific to one or the other technology.
I have rephrased the abstract so that it now clear that I am combining these two technologies, as can be seen Figure 2. It is true that the hypotheses all relate to one or the other of those two technologies, but since I see Blockchain as a main enabler for the flow of information within the PI framework, all changes are necessarily intertwined.
As the authors note, the research questions are all written at a high level of abstraction, leaving it to individual researchers to further specify and operationalize them. For this reviewer, the paper would be more interesting and useful if the authors reached for a lower level of abstraction.
I absolutely agree with you that it would be desirable to reach a lower level of abstraction and to derive research questions / hypotheses than can easily be applied to specific research problems. However, it was my intention in this paper to structure the field and to provide a fertile ground for future research. I definitely hope that other researchers will elaborate on my ideas and reach a lower level of abstraction. I explicitly mention this as a limitation / goals in the last paragraph of this paper.
Again, let me thank you for taking the time for reviewing my paper and providing useful suggestions that helped me to improve my publication. This paper was proofread by a native speaker, who is also an expert in sustainability and I hope that the final version meets your expectations.
Reviewer 3 Report
The paper “Combining Blockchain technology and the Physical Internet to Achieve Triple Bottom Line Sustainability: A Comprehensive Research Agenda for Modern Logistics and Supply Chain Management” derives several research areas and research 16 questions to investigate adoption, management as well as structural SC issues. In my opinion the content of the paper is adequate for the purposes of the journal.
Title: The title of the paper is informative. It includes important terms and the message of the article.
Keywords: Keywords are well chosen.
Abstract: The abstract describes the context and provide a general picture of the methodological approach. The main outcomes are also described.
Introduction and theoretical background: Introduction defines the focus and explains the structure of the text. Literature review prepares the reader to understand the research part of the article. A summary table comparing the contributions could support the explanation. I advise to include some articles focusing on sensor networks in SC and production (see Bányai et al. 2017 DOI: 10.4028/www.scientific.net/SSP.261.456) and complexity of supply chain solutions (see Bányai 2015 DOI: 10.1016/j.proeng.2015.01.341)if suitable.
Research agenda: TBL sustainability, a layered framework, blockchain and physical internet are discussed. Research area and research questions are extensive discussed.
Discussion, Conclusions and Future Research: Implications, limitations and future research directions are discussed. In addition, what kind of lesson emerges from the paper for managers?
Author Response
Response to Reviewer 3 comments
The paper “Combining Blockchain technology and the Physical Internet to Achieve Triple Bottom Line Sustainability: A Comprehensive Research Agenda for Modern Logistics and Supply Chain Management” derives several research areas and research 16 questions to investigate adoption, management as well as structural SC issues. In my opinion the content of the paper is adequate for the purposes of the journal.
Dear Reviewer,
thank you very much for taking the time to carefully review my paper and to provide valuable feedback. I will explain in the following sections how I have used your feedback to improve my paper.
Title: The title of the paper is informative. It includes important terms and the message of the article.
Keywords: Keywords are well chosen.
Abstract: The abstract describes the context and provide a general picture of the methodological approach. The main outcomes are also described.
Thank you for your positive assessment of the title, keywords and the abstract.
Introduction and theoretical background: Introduction defines the focus and explains the structure of the text. Literature review prepares the reader to understand the research part of the article. A summary table comparing the contributions could support the explanation. I advise to include some articles focusing on sensor networks in SC and production (see Bányai et al. 2017 DOI: 10.4028/www.scientific.net/SSP.261.456) and complexity of supply chain solutions (see Bányai 2015 DOI: 10.1016/j.proeng.2015.01.341)if suitable.
Thank you for pointing my attention to relevant research. I included both references in my paper.
Research agenda: TBL sustainability, a layered framework, blockchain and physical internet are discussed. Research area and research questions are extensive discussed.
Thank you.
Discussion, Conclusions and Future Research: Implications, limitations and future research directions are discussed. In addition, what kind of lesson emerges from the paper for managers?
My paper mainly targets academics, but I have inserted a sentence in the discussion section on how I believe that it might also benefit practitioners. In a nutshell, I believe that the latter can benefit from a better understanding of the connection between the Blockchain and the PI.
Best regards,
Horst Treiblmaier
Round 2
Reviewer 2 Report
Thank you for accommodating the recommended changes, especially improving the flow for the benefit of the reader. From my perspective it improves the paper very much.
Regarding the voice...for research papers such as yours, my preference would be to avoid the first-person active voice. This is the standard for most top-tier research journals. Some journals have sections for "Perspectives" or "Notes" which are more suitable for the use of the first-person active. I could not find the 2013 JAIS style guide that you cited. I did find another JAIS style guide, undated, but it did not reference the issue.
P.S. Minor note …. missing capital letter in title...
Author Response
Dear Reviewer,
thank you very much for taking the time to respond to my comments.
I concur that the discussion regarding the use of active voice vs. passive voice is a very subjective one. Here is the link to the style guide I was referring to:
I also present my personal opinion in a paper which was published in the latest issue of the DATA BASE for Advances in Information Systems (Vol. 50. No. 1). The paper is titled "Taking Feyerabend to the Next Level: On Linear Thinking, Indoctrination, and Academic Killer Bees". In it I elaborate on the current academic system. I discuss the issue of active vs. passive voice from page 8 onwards. I hope that you will find my comments useful.
Again, thank you for your support.
Best regards,
Horst Treiblmaier